# Dexmedetomidine Pre-Treatment of Neonatal Rats Prevents Sevoflurane-Induced Deficits in Learning and Memory in the Adult Animals

**DOI:** 10.3390/biomedicines11020391

**Published:** 2023-01-28

**Authors:** Nerea Jimenez-Tellez, Marcus Pehar, Fahad Iqbal, Alberto Casas-Ortiz, Tiffany Rice, Naweed I. Syed

**Affiliations:** 1Department of Biochemistry and Molecular Biology, University of Calgary, Calgary, AB T2N 4N1, Canada; 2Hotchkiss Brain Institute, University of Calgary, Calgary, AB T2N 4N1, Canada; 3Alberta Children’s Hospital Research Institute, University of Calgary, Calgary, AB T2N 4N1, Canada; 4Department of Anesthesiology, Perioperative and Pain Medicine, University of Calgary, Calgary, AB T2N 4N1, Canada; 5Department of Cell Biology and Anatomy, University of Calgary, Calgary, AB T2N 4N1, Canada

**Keywords:** sevoflurane, dexmedetomidine, anesthetics, learning and memory, mitochondria

## Abstract

Anesthetics have been shown to cause cytotoxicity, cell death, affect neuronal growth and connectivity in animal models; however, their effects on learning and memory remain to be fully defined. Here, we examined the effects of the inhalation anesthetic sevoflurane (SEV)—both in vivo by examining learning and memory in freely behaving animals, and in vitro using cultured neurons to assess its impact on viability, mitochondrial structure, and function. We demonstrate here that neonatal exposure to sub-clinically used concentrations of SEV results in significant, albeit subtle and previously unreported, learning and memory deficits in adult animals. These deficits involve neuronal cell death, as observed in cell culture, and are likely mediated through perturbed mitochondrial structure and function. Parenthetically, both behavioural deficits and cell death were prevented when the animals and cultured neurons were pre-treated with the anesthetic adjuvant Dexmedetomidine (DEX). Taken together, our data provide direct evidence for sevoflurane-induced cytotoxic effects at the neuronal level while perturbing learning and memory at the behavioural level. In addition, our data underscore the importance of adjuvant agents such as DEX that could potentially counter the harmful effects of commonly used anesthetic agents for better clinical outcomes.

## 1. Introduction

Brain connectivity established during development forms the basis for learning and memory in all animal species, including humans. These connectivity patterns are, in turn, contingent upon activity-dependent mechanisms that allow brain circuits underlying most behaviours—including learning and memory—to form properly. However, the rate at which the brain circuitry is orchestrated during early development, and the involvement of myriad intrinsic cell–cell signalling and external factors (trophic factors, etc.) present in the extracellular milieu makes it often difficult to decipher how neural networks are organized at the cellular or behavioural level. Moreover, it is also difficult to investigate how both intrinsic and environmental influences could potentially impact these connectivity patterns, thus leaving an animal with life-long functional deficits. For example, in the retinotectal pathway, if the sensory input is blocked during early development while brain maps are being put together, vision will likely be compromised for life [1]. Similarly, young animals devoid of an enriched and nurturing environment exhibit learning and memory deficits throughout their adult life [2]. An early mitigating intervention, on the other hand, by either removing the potential interrupters of proper circuit formation or by offering the young animals an enriched environment could reverse the situation through the invocation of synaptic plasticity [2,3,4].

There exists ample evidence to support the notion that among the most critical factors mediating brain connectivity are the activity-dependent mechanisms [5,6,7], the perturbation of which during a critical period of brain development renders synaptic connectivity compromised for life. Several terms have been coined to describe such a concept; for instance, the cells that “fire together, wire together” [8]. Selective blockade of electrical activity in specific brain regions to determine their impact on any given behaviour is, however, a formidable task. The challenge is even more daunting for distributed neuronal networks that underlie learning and memory. In most instances, the potential extrinsic interrupters of brain network connectivity could be avoided, or the shortcomings managed through synaptic plasticity involving compensatory mechanisms [9]. However, there are some circumstances when either blocking or suppressing the electrical activity of the entire brain becomes unavoidable. Such situations arise when young babies or children need to undergo surgical interventions requiring anesthesia.

Anesthetics are used as a general practice for many clinical procedures, and their efficacy and safety have improved significantly over the years culminating in newer and safer compounds. Notwithstanding their safe practical use, evidence is beginning to emerge demonstrating the cytotoxic effects of a number of clinically used anesthetics, especially when tested in animal models [10]; however, the evidence for their impact on learning and memory in humans remains controversial [11,12,13,14]. Given that the anesthetic-induced state of anesthesia involves blocking both neuronal impulse conduction and synaptic transmission, it thus follows that an early anesthetic exposure of young children with developing brain circuits will also prevent neuronal firing, which may in turn perturb or interfere with activity-dependent mechanisms underlying proper network formation. However, this postulate cannot be tested experimentally in humans, and the evidence in most animal models remains inconclusive.

One of the most commonly used anesthetics in pediatric medicine is sevoflurane. This compound is less irritating than other volatile anesthetic agents when inhaled and has faster induction and quicker recovery properties [15]. Sevoflurane is thought to induce anesthesia via γ-aminobutyric acid A (GABAA) receptors, which are responsible for the suppression of fast synaptic transmission [16], and comprise the major inhibitory system in the adult central nervous system (CNS) [17]. However, during early development, GABAA receptors serve an opposite function to that of the mature brain. Specifically, during development, the GABA receptors conduct excitatory currents due to chloride transporter reversal when bound to their ligands [18]. This reversal of function during early life is thought to play an important role during brain development, which in turn impacts network assembly through the involvement of synaptic activity [18]. Even though it is unknown when this switchover from excitatory to inhibitory function occurs in humans, there is some evidence to suggest that it may begin at birth and be completed around two weeks thereafter [19]. It is important to note that this time window also represents a period of major programmed cell death—concomitant with the refinement of brain circuits, thus giving rise to the final patterns of brain connectivity [20].

Several studies have shown that sevoflurane exposure of young neurons induces cell death, possibly involving increased mitochondrial apoptotic caspase activity [21], reactive oxygen species (ROS) production [22], or decreased Bcl function [23]. Furthermore, several animal studies have described that either higher concentrations (2.5–3.5%) [24,25,26] or repeated sevoflurane exposure during early developmental stages [24,25] may induce detrimental effects on learning and memory throughout adulthood. In contrast, other studies have demonstrated that sevoflurane exposure does not affect cognition [27,28] or even improve it [29]. Similar controversies and discrepancies also exist in the case of human studies [12,13,14]. It is therefore important, on the one hand, to resolve these issues both at the cellular and behavioural level and, on the other hand, to decipher the precise mechanisms underlying potential sevoflurane-induced toxicity in the developing brain.

Recent studies have shown a newer anesthetic agent, Dexmedetomidine (DEX) to exhibit either little or no cytotoxicity when used in humans [30] or tested in animal models [31]. Moreover, DEX has also been touted as a potential neuroprotective agent as it mitigates cytotoxicity induced by other agents [32,33,34]. DEX exerts its anesthetic action as an α2-adrenergic receptor agonist [35], thus mimicking the activation of an endogenous sleep pathway [36], and as such, significantly differs from other commonly used anesthetics targeting NMDA, GABA, or other transmitter receptors [37]. Studies have suggested that this anesthetic is safe and neuroprotective both at the cellular and behavioural levels when used either alone [31] or as an adjuvant [34,38,39]. Additional studies are, however, needed to further confirm and validate these findings.

To investigate the potential effects of neonatal sevoflurane exposure on brain development, we have established an in vivo rat model and an analogous in vitro model using primary cultured neurons. We used P7 neonatal pups which were exposed to subclinical concentrations of sevoflurane (1.6%) for one hour for either one or two days (P7 and P8) in order to understand the long-term neurological changes indicative of compromised learning and memory in adult animals. In parallel, we used the in vitro model to understand these long-term impairments at the cellular level by assessing the effects on viability and mitochondrial structure and function. We further employed both in vivo and in vitro models to understand if DEX pre-treatment could prevent sevoflurane-induced cytotoxicity and behavioural deficits.

## 2. Materials and Methods

### 2.1. Experimental Animals and Cell Culture

Sprague-Dawley pregnant rats (strain code 400) were obtained from a commercial breeder (Charles River Laboratories, Senneville, QC, Canada). Pregnancy was monitored from the day of arrival (E16.5) until the date of birth (post-natal day zero—P0). The total number of pups used for these experiments was 66, from 9 litters from different breeding groups.

P0 wild-type Sprague-Dawley rats were used for all culture experiments. Rats were housed in a conventional room where the temperature was set at 25 °C on a 12 h light/dark cycle from 7 am to 7 pm and fed *ad libitum.* Sedation was achieved via ice-induced hypothermia, where the P0 animals were wrapped in tissue paper and placed in an ice-filled container for 7 min [31,40]. Following the loss of movement and prior to them regaining consciousness, decapitation was performed to achieve euthanasia in all pups. Cortical tissue was collected immediately after decapitation.

### 2.2. Animal Dexmedetomidine Treatment

P7 rat pups of equally distributed sexes were injected with 25 µg/kg of dexmedetomidine (DEX) (Dexmedetomidine hydrochloride, Sigma-Aldrich cat. SML0956) subcutaneously either once (1×) or twice (2×), (24h after the first injection), using a 0.1 mL volume. Control animals were injected with an equivalent volume of saline (sodium chloride 0.9% Braun, Mississauga, ON, Canada). Body temperature was maintained with the use of a heated blanket and gloves filled with warm water replaced every 20 min [31]. The respiratory rate of the animal was monitored by counting breaths per minute and checked every 10 min following the first injection over the subsequent 30 min, to confirm the pup’s health. Furthermore, oxygen saturation (SpO_2_) was monitored every 30 min following each injection for a period of an hour to further ensure the pup’s health and reaction to the compound (Nonin Medical, Minneapolis, MN, USA).

### 2.3. Animal Sevoflurane Exposure

After a 1 h pre-treatment either with DEX or saline, the pups were placed in an anesthetic chamber where sevoflurane (Baxter Corporation, cat. CA2L9117) was delivered as a sevoflurane-medical air–oxygen gas mixture (75% O_2_) vaporized using a GE Datex-Ohmeda Aestiva/5 Anesthesia Machine vaporizer, and the concentration was monitored with a GE Healthcare Gas Analyzer. The sevoflurane concentration used was 1.6% for 1 h at P7 (1×) and/or P8 (2×). Breaths per minute were monitored every ten minutes while SpO_2_ was monitored before and after the anesthetic exposure. Animals were monitored for righting reflex following anesthetic exposure and upon recovery returned to their mothers.

Control animals were injected with a volume of saline of 0.1 mL and animals were monitored for breaths and SpO_2_ for the following hour. These animals were isolated in a separate box with a heat source while the sevoflurane-exposed animals were exposed to their anesthetic. All animals were kept separated from the mother for approximately 2 h and returned together once all of the animals recovered the righting reflexes.

Each pup’s ear was notched at P14, and pups were weaned at P21 in cages of 2 to 3 animals, separated by sexes. Animals were kept undisturbed—other than routine bedding changes and restocking food and water—until P60 and then subjected to various behavioural tests.

**Note:** For proper and in-depth behavioural assessment, we either introduced or adopted several qualitative parameters to describe anomalies in an animal’s behavioural repertoire that are generally not reported in most studies. Specifically, we have included and introduced terms such as: “erratic” [41,42], “hesitant” [43,44], “unsure”, “undecisive”, exploratory” [44,45], “anxious” [43,44], and “reluctance” as an animal’s behavioural responses that were incongruent with those of the untreated animals.

### 2.4. Morris Water Maze

We performed the test as previously described [31] with the animals placed in a 1.8 m diameter pool filled with water and left to find a hidden platform using visual cues placed on the walls of the room. The test was performed over 5 consecutive days and latency (time taken) to reach the platform and distance swum were measured using the automated software, ANY-Maze (Stoelting Co, Wood Dale, IL, USA). Latency and distance were used as the learning and memory parameters characterized over the period of days 1 and 5.

### 2.5. Novel Object Recognition

This test was carried out as previously described [31]. The animals underwent a habituation phase that lasted 15 min in an empty box for 2 consecutive days. On the third and fourth days, the animals were left to interact with two identical objects for 5 min during the familiarization phase, and then, removed from the box for a 5 min retention interval. During the testing phase, the animal was left to interact with a novel object that replaced one of the familiar objects (Day 3) or with a familiar object placed in a different location in the box (Day 4). Time spent interacting with the objects was monitored using ANY-Maze (Stoelting Co, Wood Dale, IL, USA).

### 2.6. Open Field

During the previously described habituation process, where each animal was placed in an empty box with a bottom of bedding and left to interact in this new environment for 15 min, the animal was observed for its locomotor activity, anxiety, and willingness to explore a new environment using open field testing [46]. The field was divided into 4 different areas: corners, walls, inner area, and center for quantification. The time spent in each area, entries made to each area, total distance travelled, track paths, and time that the rat was immobile were monitored.

### 2.7. Cell Culture Dexmedetomidine Treatment

Frontal cortices from Sprague-Dawley rats were isolated at P0 and cultured as described previously [40]. Some cultures were treated with various concentrations of DEX (0.05 µM, 0.1 µM, 1 µM, 2.5 µM, 5 µM, or 10 µM) dissolved in culture media, whereas the controls only had culture media [31]. The concentrations used here were chosen using adose–response assay based on previous literature and also clinical equivalents [32,47]. The cells were incubated at 37 °C and 5% CO_2_ for 1 h before sevoflurane exposure.

### 2.8. Cell Culture Sevoflurane Exposure

After 1 h incubation at 37 °C and 5% CO_2_, the cells were exposed to approximately 0.5 minimum alveolar concentration (MAC) equivalent (for neonates) of sevoflurane (1.6%) in an airtight modular incubator chamber (Billups-Rothenberg, San Diego, CA, USA) for one hour. Sevoflurane-medical air–oxygen gas mixtures (21% O_2_) were then vaporized using a Datex-Ohmeda Aestiva/5 vaporizer and concentrations were monitored with a GE Healthcare Gas Analyzer. Controls were exposed to medical air only. After 1 h sevoflurane-medical air–oxygen mixture or just medical air exposure, the neurons were placed back and maintained in an incubator (37 °C, 5% CO_2_) until use.

### 2.9. Cell Viability Assay

The effects of anesthetics on neuronal viability were tested on 1, 3, and 7 days post-culture with the LIVE⁄DEAD Viability/Cytotoxicity Kit (Thermo Fisher Scientific Invitrogen, Waltham, MA, USA, cat. L3224). This time point was selected to minimize any glial proliferation that could otherwise mask changes in the viability of developing neurons. Specifically, the cells were exposed to calcein-AM (green, live cells) and ethidium homodimer-1 (red, dead cells) dyes at room temperature for 15 min and imaged using a Zeiss Axio Observer Z1 microscope (Zeiss Corp., Ottawa, ON, Canada) with a 10X objective [31]. The percentage of alive cells was automatically counted using the Cell Counter plugin in ImageJ with the same thresholds for all the treatments.

### 2.10. Live-Cell Fluorescent Imaging and Confocal Microscopy

To assess the impact of DEX on the morphological integrity of the mitochondria, we used a protocol previously established in the lab [31] where the mitochondrial morphology was evaluated on day 4 post-culture. Fluorescence images were taken with an Olympus SD-OSR spinning-disk confocal microscope (100×/1.49 oil) with a mounted incubator system (Olympus Corp., Richmond Hill, ON, Canada) and all imaging parameters were kept identical for each dish. Mitochondrial morphology was quantified as previously described [40].

### 2.11. Reactive Oxygen Species (ROS) Production Using Flow-Cytometry

ROS production was quantified over time (Days 1, 2, 3, 4, 7, and 10 following experimental treatment) as previously described [31] using flow cytometry. For the gating criteria we first used SSC-A (side scatter/relative granularity) vs. FSC-A (forward scatter area/relative size) to separate the single cells, second, we used FSC-A vs. FSC-H (forward scatter height) to separate the different potential brain-derived population by size (keeping the neuron gate) and third, we used FSC-A X FITC-A (calcein positive) to select only the cells that were alive. The apparatus used for cell sorting was BD LSR II (BD Biosciences, USA) and the software used for data analysis was FlowJo (BD Biosciences, San Jose, CA, USA).

### 2.12. Statistical Analysis

All samples were designated randomly, and the experiments were conducted in a single-blinded fashion. Specifically, the observer was uninformed of the experimental conditions. Statistical significance tests were performed with GraphPad Prism 8. One-way ANOVA was used followed by Dunnett’s multiple comparisons in order to analyze single independent variants and two-way ANOVA was used to compare multiple groups with two independent variants, followed by Tukey’s multiple comparisons tests for post hoc comparisons. Differences between the means of the two conditions were tested using the two-sided Student’s t-test with Welch’s correction. Differences between data were considered significant if appropriate post hoc statistical tests resulted in *p* ≤ 0.05 [31].

## 3. Results

### 3.1. Sevoflurane Exposure of Rat Pups Did Not Affect Locomotor Skills, but the Exploratory Nature of Their Behaviour Was Altered

To determine if sevoflurane exposure of the rats during the early stages of life may affect some aspects of their locomotor behaviour and learning and memory during adulthood, the pups were exposed to sevoflurane 1.6% for 1 h at P7 for the single exposure group (1×) or to sevoflurane 1.6% for 1 h at both P7 and P8 (24 h apart) for the double exposure (2×).

When the animals reached sexual maturity (P60), they were subjected to behavioural testing to assess the long-term impact of sevoflurane on their learning and memory (Figure 1A).

To assess the level of anxiety, locomotor skills and “willingness” to explore their new environment, the animals underwent open field testing. Total distance travelled within the box (Figure 2C) and immobile ratio (Figure 2D) were compared between the groups using one-way ANOVA statistical tests.

There were no significant differences in either of these parameters between the sevoflurane groups (1× or 2×) compared to the control group. Time spent in each area (Figure 2A) and the number of entries made into each area (Figure 2B) were also assessed and analyzed usingt two-way ANOVA. In all groups, there was a preference for the subsequent areas as follows: corners > walls > inner areas > centre, indicative of a regular pattern of “anxiety-like” behaviour where the animals sought to prefer the most sheltered areas [48]. There was, however, a significant difference in the sevoflurane 1.6% 1 h (2×) group where the animals spent a significantly increased length of time exploring the corners (mean = 571.6, SEM = 59.75, *p* = 0.03, *n* = 6) compared to their control counterparts (mean = 480.2, SEM = 17.78, *n* = 13). However, when looking at the number of entries per area (Figure 2B), there were no significant differences between the groups.

Finally, we created heatmaps to decipher the preferences made by each group in the box. We observed that the control animals evenly distributed their time through all four corners, spending the least amount of time in the top left corner (Figure 2I), similar to what we saw in the sevoflurane 1.6% 1 h (1×) group (Figure 2E), whereas the sevoflurane 1.6% 1 h (2×) group mainly confined itself to the bottom corners (Figure 2G). It is important to note that the controls did show a trend where they spent more time exploring the walls (as shown by the lighter blue areas in the wall areas), unlike the sevoflurane-exposed animals that were more prone to confine themselves to the corners.

The above data thus demonstrate that while sevoflurane did not affect the locomotor behaviour as all animals spent equal lengths of time exploring the box, the rats with repeated exposure to this anesthetic appeared “reluctant” [43,44] to explore other quadrants and preferred to confine themselves to the corners instead.

### 3.2. Sevoflurane Exposure of the Neonates Subtly Altered Several Important Aspects of Their Spatial Memory

To test whether sevoflurane exposure had any long-term effects on spatial learning and memory, we used the Morris Water Maze (MWM) test. Specifically, the animals were placed in a pool and left to find a hidden platform using distinct spatial cues located on the walls of the room over a period of five consecutive days. The trajectory followed by the animals while swimming in the pool was monitored during each trial. Representative images of three of the conditions (control, Sev 1×, and Sev 2×) are represented in Figure 3 for each quadrant for days 1 and 5 (Figure 3B,C,E,F). The average latency (time needed to reach the platform) (Figure 3A) and distance swum (Figure 3D) were measured for each trial. Two-way ANOVA was used to compare the average latency and distance between days 1 to 5 for each group. For the latency, we saw a significant decrease in the time required to reach the platform when comparing days 1 and 5 for all sevoflurane-exposed groups: Sevoflurane 1.6% 1 h (1×) (Day 1: mean = 47.08, SEM = 3.441 *n* = 9 and Day 5: mean = 21.96, SEM = 2.891, *p* = 0.0004, *n* = 9), Sevoflurane 1.6% 1 h (2×) (Day 1: mean = 38.76, SEM = 4.2, *n* = 9 and Day 5: mean = 17.74, SEM = 2.288, *p* = 0.0087, *n* = 9), as well as the control group (Day 1: mean = 46.63, SEM = 2.728, *n* = 13 and Day 5: mean = 21.19, SEM = 2.806, *p* < 0.0001, *n* = 13), suggesting that spatial learning had occurred in all groups. Furthermore, when comparing the differences between the control group and both sevoflurane groups both on days 1 and day 5, there were no statistically significant differences in the latency among the three groups, further suggesting that the exposure to sevoflurane did not affect the latency to reach the platform.

With regards to the distance travelled by the animals (Figure 3D), there was no significant improvement between days 1 and 5 in the sevoflurane 1.6% 1 h (1×) group (Day 1: mean = 19.89, SEM = 4.138, *n* = 9 and Day 5: mean = 7.652, SEM = 2.289, *p* = 0.6568, *n* = 9) compared to the controls (Day 1: mean = 20.14, SEM = 3.356, *n* = 13 and Day 5: mean = 6.328, SEM = 1.191, *p* = 0.0005, *n* = 13), but it was significant for the sevoflurane 1.6% 1 h (2×) group (Day 1: mean = 37.73, SEM = 5.41, *n* = 9 and Day 5: mean = 6.607, SEM = 1.322, *p* < 0.0001, *n* = 9). The significance resides in the fact that, initially, the distance swam by the Sev 2X group (37.73 m ± 5.41), was higher as the animals swam longer distances than the control (28.74 m ± 5.739) and Sev 1X (19.89 m ± 4.138) groups on day 1.

A remarkable difference was observed when the animal’s swimming trajectory was carefully monitored and recorded during this observation period. A careful analysis of the tracings of sevoflurane-exposed animals’ swimming trajectories revealed an interesting pattern where although all the animals were able to reach the platform with similar latency on day 5 (*p* > 0.9999), the rats that had been exposed to sevoflurane did not follow a direct trajectory to reach their platform. Rather, these animals exhibited an erratic swim pattern—swimming faster in anticipation of locating their platform. Specifically, the animals appeared “anxious” to find the platform (intense swimming behaviour), albeit not knowing initially where it might have been located. In contrast, the control animals had no difficulty remembering where the potential platform would likely be, and they swam directly toward their desired target.

The data presented above suggest that even though the animals did learn to recognize and reach the platform over the 5-day period, rats that were exposed to sevoflurane earlier in life were nevertheless more “unsure” as to where the potential target might have been located. It is important to note that this aspect of their learning and memory behavioural repertoire did not significantly improve or change over a period of 5 consecutive days, thus suggesting it to be a potentially permanent impairment.

### 3.3. Postnatal Sevoflurane Exposure Impaired Recognition Memory

To understand the long-term effect of early life exposure to sevoflurane on hippocampal-dependent recognition memory, we used the NORT experimental paradigm and measured the discrimination index [DI = TN/(TN + TF)] where TN = time spent while exploring the new object or the object placed at a different location and TF = time spent exploring the familiar object, and the relative time spent exploring the objects: familiarization (TN + TF)/300; testing (TN + TF)/180. In all of the sevoflurane-exposed groups, when a novel object was introduced, there was no significant increase in the time that the animals spent interacting with the novel object (Sev 1.6% 1 h (1×): mean = 0.7077, SEM = 0.0761, *p* = 0.7888, *n* = 9, Sev 1.6% 1 h (2×): mean = 0.694, SEM = 0.0427, *p* = 0.092, *n* = 8) as compared to the familiarization phase (Sev 1.6% 1 h (1×): mean = 0.5759, SEM = 0.0433, *n* = 9, Sev 1.6% 1 h (2×): mean = 0.465, SEM = 0.0339, *n* = 8) which had occurred in the saline controls (Testing phase: Saline: mean = 0.6939, SEM = 0.0496, *p* = 0.0157, *n* = 12; Familiarization phase: Saline: mean = 0.4698, SEM = 0.0384, *n* = 12) (Figure 4A). With regards to the relative time interacting with both objects for the two phases, there were no significant differences between the groups (Figure 4B), ruling out the possibility that the overall exploration time might have been responsible for the differences observed when introducing the novel object.

On the other hand, when one of the familiar objects was placed at a different position in the box, in both of the sevoflurane-exposed groups, the discrimination index was significantly higher in the testing phase (Sev 1.6% 1 h (1×): mean = 0.6438, SEM = 0.0246, *p* = 0.0304, *n* = 9; Sev 1.6% 1 h (2×): mean = 0.6851, SEM = 0.0361, *p* < 0.0001, *n* = 9) compared to the familiarization phase (Sev 1.6% 1 h (1×): mean = 0.4487, SEM = 0.0429, *n* = 9; Sev 1.6% 1 h (2×): mean = 0.3786, SEM = 0.0334, *n* = 9). This was also observed in the controls (Testing phase: Saline: mean = 0.6554, SEM = 0.0487, *p* < 0.0001, *n* = 12, and Familiarization phase: Saline: mean = 0.3938, SEM = 0.0276, *n* = 12) (Figure 4C), highlighting the fact that the animals did not have their spatial recognition compromised after sevoflurane exposure during early stages of their life. Again, the relative time spent exploring the object was not significantly different between the groups (Figure 4D).

From these behavioural experiments, we deduced that sevoflurane exposure during early life exerted a permanent, long-term detrimental effect on recognition memory but not on spatial memory.

### 3.4. DEX Pre-Treatment Prevented the Animals from Sevoflurane-Induced Deficit in Learning and Memory

We next sought to determine if DEX could promote a rescue effect on sevoflurane’s detrimental effects on cognition. To do so, we administered DEX as a pre-treatment in the P7 pups prior to their exposure to sevoflurane. We have previously shown that subcutaneous administration of 25 μg/kg DEX either once (P7) or twice (P8) did not impact long-term spatial or recognition memory, thus highlighting it's safety when given subcutaneously [31].

Animals were subjected to behavioural tests to assess the impact of DEX pre-treatment prior to their exposure to sevoflurane (as above). For open field testing, total distance travelled within the box (Figure 2C) and immobile ratio (Figure 2D), were compared between the groups using one-way ANOVA. We did not observe any significant differences in any of these parameters between the different groups. Time spent in each area (Figure 2A) and the number of entries made to each area (Figure 2B) were also assessed using two-way ANOVA. In all groups, as seen before, there was a predilection for the subsequent areas as follows: corners > walls > inner areas > centre, where the animals preferentially stayed in the most secluded areas. With regards to the animals pre-treated with DEX and later exposed to sevoflurane (2×), they did not exhibit any increased time spent in the corners (mean = 511.0, SEM = 20.63, *p* = 0.9995, *n* = 6), in contrast to their sevoflurane-only counterparts (mean = 571.6, SEM = 59.75, *p* = 0.03, *n* = 6) (Figure 2A).

With regards to the heatmaps, the animals pre-treated with DEX and subsequently exposed to sevoflurane (2×) did, however, explore all of the corners (Figure 2H) compared to their sevoflurane-only (2×) counterparts, which only explored the two bottom corners (Figure 2G). The DEX-pre-treated animals exposed to sevoflurane (1×) spent less time in the corners but more time along the wall (Figure 2F) showing a less anxious and more exploratory phenotype than the sevoflurane-only (1×) counterparts (Figure 2E), similar to the saline control group (Figure 2I).

In summary, the subcutaneous pre-treatment with DEX did not seem to significantly affect locomotor behaviour, but it decreased the “reluctance” and “hesitation” behaviours, and thus these animals exhibited a more exploratory phenotype [49].

### 3.5. DEX Pre-Treatment Prevented the Sevoflurane-Induced Deficit in Spatial Memory

We used the Morris Water Maze to test whether DEX pre-treatment could mitigate sevoflurane-induced effects on spatial memory when animals were tested several weeks after neonatal exposure to this inhalation anesthetic. The average latency (Figure 3A) and distance swum (Figure 3D) were measured for each trial. In both groups pre-treated with subcutaneous DEX (Sev + Dex 1× and 2×), we saw a significant decrease in the latency (Sev + dex 1.6% 1 h (1×): Day 1: mean = 46.08, SEM = 2.531, *n* = 9 and Day 5: mean = 19.56, SEM = 2.546, *p* = 0.0002, *n* = 9 and Sev + Dex 1.6% 1 h (2×): Day 1: mean = 39.61, SEM = 3.793, *n* = 9 and Day 5: mean = 17.16, SEM = 2.432, *p* = 0.0057, *n* = 9), suggesting that the pre-treated animals did not differ from the saline-injected animals; that is, there was neither any improvement nor deterioration of spatial memory. On the other hand, as shown for the Sev 1× animals, the distance travelled did not significantly decrease over time for the DEX pre-treated counterparts (Sev + Dex 1.6% 1 h (1×): Day 1: mean = 22.27, SEM = 4.763, *n* = 9 and Day 5: mean = 8.877, SEM = 3.088, *p* = 0.8149, *n* = 9). On the contrary, in the Sev 2× and Sev + Dex 2× groups, the distance significantly improved (Sev + Dex 1.6% 1 h (2×): Day 1: mean = 37.42, SEM = 8.523, *n* = 9 and Day 5: mean = 5.562, SEM = 1.562, *p* < 0.0001, *n* = 9 and Sev + Dex 1.6% 1 h (2×): Day 1: mean = 37.73, SEM = 5.410, *n* = 9 and Day 5: mean = 6.607, SEM = 1.322, *p* < 0.0001, *n* = 9), due to the fact that they parted from an initial erratic and longer swimming trajectory in comparison to the other animal groups.

Interestingly, when looking at the swimming patterns, we observed that similar to the controls, the trajectories of the DEX pre-treated animals (Sev + Dex 1× and 2×) seemed more straightforward, and the animals were more “decisive” and less “unsure”, as they could tell where the platform was located (Figure 3B,C,E,F).

### 3.6. DEX Pre-Treatment Prevented the Sevoflurane-Induced Deficits in Recognition Memory

As shown above, sevoflurane exposure of the neonatal animals, irrespective of the number of exposures, compromised recognition memory. We tested whether DEX pre-treatment could mitigate the sevoflurane-induced effects on recognition memory. The animals were injected with DEX as above prior to their sevoflurane exposure. We observed that subcutaneous DEX pre-treatment did significantly improve recognition memory (Testing phase: Sev + Dex 1.6% 1 h (1×): mean = 0.7907, SEM = 0.06146, *p* = 0.0011, *n* = 9; Sev + Dex 1.6% 1 h (2×): mean = 0.7029, SEM = 0.08425, *p* = 0.0274, *n* = 9) (Figure 4A,B) compared to the sevoflurane-only exposed animals (Sev 1.6% 1 h (1×): mean = 0.7077, SEM = 0.0761, *p* = 0.7888, *n* = 9, Sev 1.6% 1 h (2×): mean = 0.694, SEM = 0.0427, *p* = 0.092, *n* = 8). These data demonstrate that indeed the behavioural phenotype significantly improved to control levels when the animals were pretreated with DEX prior to sevoflurane exposure and that this approach helped prevent the long-term deterioration of recognition memory caused by sevoflurane.

When looking at the discrimination ratio as the familiar object was placed at a different location in the box, we observed significant differences between the familiarization and the testing phase for all experimental groups, indicating that none of the experimental groups had problems recognizing the change of location of the object (Figure 4C,D).

### 3.7. Sevoflurane Mediated Effects on Learning and Memory May Involve Cell Death

As sevoflurane has been shown to exhibit cytotoxicity at the cellular levels [32,33,50], we next sought to determine whether its observed effects on learning and memory may have involved cell death or the perturbation of neuronal viability. To address this question, we exposed isolated cortical neurons to 0.5 minimum alveolar concentration (MAC) equivalent of sevoflurane (1.6%) vaporized into a mixture of medical air and oxygen in an airtight modular incubator chamber for one hour. We then quantified the percentage of alive–dead cells in the sample on days 1, 3, and 7 after sevoflurane exposure (Figure 5A–D).

We found a higher percentage of dead cells in the sevoflurane-exposed dishes (Day 1: mean = 37.66, SEM = 0.2028, *p* < 0.0001, *n* = 3 and Day 3: mean = 49.25, SEM = 2.988, *p* < 0.0001, *n* = 3) as compared with their control counterparts (Day 1: mean = 72.87, SEM = 4.515, *n* = 3 and Day 3: mean = 82.3, SEM = 2.928, *n* = 3) both on day 1 (Figure 5A,B) and 3 (Figure 5C,D). On day 7, however, there were no significant differences between the controls (mean = 78.33, SEM = 0.8819, *n* = 3) and the ones treated with sevoflurane (mean = 73.47, SEM = 1.316, *p* = 0.7670, *n* = 3) (Figure 5E). These results suggest that the exposure to sevoflurane acts in a short-term manner and that the cytotoxicity fades away over time in the cortical cells, presumably because the anesthetic is no longer in contact with the cells.

### 3.8. Sevoflurane-Induced Cytotoxic Effects on Neuronal Cultures Involved Mitochondrial Network Fragmentation, Which Was Independent of ROS Production

Having shown that sevoflurane exposure resulted in increased cell death, we next sought to determine the mechanism of toxicity. Former studies have reported mitochondria as potential targets for anesthetic-induced toxicity [10]. We thus examined whether sevoflurane-induced cell death may have also affected this organelle. To determine the sevoflurane-induced effects on mitochondrial morphology and the ensuing ROS production, we performed live-cell imaging using MitoTracker Green to assess mitochondrial morphology and flow cytometry to measure ROS production as previously described [31,40]. We observed that the sevoflurane-exposed cells exhibited significantly more fragmented mitochondrial networks on day 4 post-exposure (Figure 6A) (mean = 30.92%, SEM = 1.273 fragmented mitochondria, *p* = 0.0496, *n* = 8), compared to the control cells (mean = 20.03%, SEM = 3.941 fragmented mitochondria, *n* = 8), with reduced intermediate morphologies (mean = 55.78%, SEM = 2.448 intermediate mitochondria, *p* = 0.0013, *n* = 8), compared to the control cells (mean = 71.45%, SEM = 3.439 intermediate mitochondria, *n* = 8), while the fused fraction was not significantly different (mean = 13.3%, SEM = 1.752 fused mitochondria, *p* = 0.7775, *n* = 8), compared to the control cells (mean = 8.519%, SEM = 1.357 fused mitochondria, *n* = 8). These data thus demonstrate that sevoflurane alters mitochondrial morphology away from a healthy state and towards a less functionally effective conformation.

To test whether sevoflurane-induced mitochondrial fragmentation resulted in greater Reactive Oxygen Species (ROS) production, which in turn may underlie the observed cell death of cultured neurons, we used a flow-cytometry approach as previously described [31]. Specifically, we measured ROS production on days 1, 2, 3, 4, 7, and 10 after sevoflurane exposure and compared those data with the untreated control cultures (Figure 6B,C). Both sevoflurane and control cells showed a distinct ROS production profile over time as indicated by fluorescent intensity, which was not statistically different for any of the days analyzed between both groups (Figure 6D). These data demonstrate that notwithstanding its effects on mitochondrial morphology, neuronal sevoflurane exposure does not affect ROS production and as such, this may not be the underlying cause for the observed anesthetic-induced cell death.

### 3.9. DEX Pre-Treatment Prevented Sevoflurane-Induced Cell Death in a Dose-Dependent Manner

Having demonstrated that clinically used sevoflurane concentrations caused significant cell death, we next asked the question whether another adjuvant anesthetic DEX (previously shown by us and others to exhibit no cytotoxicity [21,31]) could rescue neurons from the sevoflurane-induced cell death. To test this possibility, we explored various different concentrations of DEX (0.05 µM, 0.1 µM, 1 µM, 2.5 µM, 5 µM, or 10 µM). The cultured neurons were pre-treated with different concentrations of DEX for one hour prior to their exposure to sevoflurane. Remarkably, we found that on day 3 post-exposure, pre-treatment of cells with concentrations of DEX below 10 µM reversed sevoflurane-induced cytotoxicity in a dose-dependent manner, with 1 µM DEX being the concentration exhibiting the most remarkable recovery (Sev + Dex: mean = 81.98, SEM = 3.382, *p* = 0.0048, *n* = 3: Sev: mean = 49.25, SEM = 2.988, *n* = 3) (Figure 7A,B).

### 3.10. DEX Pre-Treatment Prevented Sevoflurane-Induced Fragmented Mitochondrial Networks and Diminished ROS Production

Having shown that DEX pre-treatment successfully prevented neurons from increased cell death, we next asked if this increased survivability also involved mitochondrial health. As previously shown, DEX exposure at a concentration of 1 μM promoted a hyperfused mitochondrial network in a time-dependent manner [31]. Therefore, we proceeded to pre-treat the cells with DEX 1 μM before exposing them to sevoflurane (1.6% for 1 h) and then assessed the mitochondrial morphology on day 4 post-exposure. Cells pre-treated with DEX showed hyperfused mitochondrial networks (mean = 29.79%, SEM = 4.74 fused mitochondria, *p* = 0.0048, *n* = 7), compared to the control cells (untreated cells) (mean = 8.519%, SEM = 1.357 fused mitochondria, *n* = 8) (Figure 7C), consistent with what we had reported previously [31]. Remarkably, when sevoflurane-exposed cells were pre-treated with DEX, not only did the fused fraction significantly increase, but this pre-treatment also fully reversed the fragmented phenotype (mean = 11.29%, SEM = 2.826 fragmented mitochondria, *p* = 0.0024, *n* = 7), that was otherwise observed in the sevoflurane-exposed cultures (mean = 30.92%, SEM = 1.273 fragmented mitochondria, *n* = 8) (Figure 7D).

Even though we did not observe significant changes in the ROS production in the sevoflurane-exposed cultures, we still tested the effect of DEX pre-treatment on this parameter. Interestingly, we saw a significant difference in the mean fluorescence intensity (MFI) on day 7 (Figure 7E,F) between controls (untreated cells) and DEX-pre-treated cells (control: D7 mean = 1613, SEM = 212.0, *n* = 4 and Sev + Dex: D7 mean = 1227, SEM = 131.9, *p* = 0.0221, *n* = 4). These data can be corroborated with the results shown in the previous assay where we saw an increase in the hyperfused mitochondrial fraction in the DEX-pre-treated cells suggesting a possible neuroprotective mechanism for DEX involving mitochondrial health—both at the morphological and ROS production levels.

## 4. Discussion

Notwithstanding recent efforts to develop safer and more effective anesthetic agents, most compounds in clinical use continue to present some long-term side effect concerns [10]. Whereas both the nature and the magnitude of these effects on the human brain remain unknown, ample evidence from animal studies points towards potential harmful effects, especially cytotoxicity that has been observed both in vitro and in vivo [11,12,13]. Here, we provided direct evidence that sevoflurane, which is one of the most commonly used anesthetics in pediatric patients, induces cell death in young cell cultures and learning and memory deficits in adult animals exposed to this anesthetic postnatally. We also reported some subtle, albeit important and long-term changes in an animal’s learning and memory repertoire that have not been reported previously. Furthermore, we demonstrated that these effects likely involved perturbation of neuronal circuit formation and mitochondrial fragmentation. In addition, we found that sevoflurane-induced effects on cell death and learning and memory could be prevented by DEX pre-treatment. Together with our previously published studies [31], where we demonstrated that DEX not only promoted neuronal growth but also mitochondrial health, our data further underscores the importance of this adjuvant as a potential neuroprotective agent.

Our in vitro data demonstrated that the exposure of cortical neurons extracted from newborn rats to half minimum alveolar concentration (MAC) (1.6%) sevoflurane for 1 h was sufficient to cause detrimental effects on neuronal health with the neurons exhibiting significantly increased cell death. This effect on neuronal viability was most evident on day 1 but subsequently attenuated over time, with there being either little or no effect on day 7. While this study is the first to examine the long-term effects of sevoflurane on cultured neurons, our data also demonstrated neuronal vulnerability to this agent specifically during the early stages of their morphogenesis.

In this study, not only did we monitor the effects of sevoflurane on cellular viability, but we also examined its impact on mitochondrial health by analyzing their morphology at day 4 post-exposure, while monitoring ROS production over an extended time period. Even though we did not find any significant differences in the ROS production profile as compared with the control cells, we did nevertheless notice changes in the mitochondrial morphology. Specifically, we discovered that the cells exposed to sevoflurane had significantly more fragmented networks compared to their control counterparts. Previous studies have provided differing accounts regarding the effects of sevoflurane on mitochondrial morphology. For instance, some studies have reported smaller mitochondrial diameters in rats exposed at early stages in life and when examined during adulthood [51] while others have shown no change in mitochondrial morphology after sevoflurane exposure [52]. In another study, an increase in the fused fraction was noted when cardiac cultures were exposed to 2.4% sevoflurane for a very brief time after hypoxia/reoxygenation injury [53]. Similarly, ROS production has previously been reported in the literature in response to sevoflurane exposure, although the concentration and the exposure time resulting in drastic increases in ROS production were far greater than what we used in the present study [51].

Different researchers have studied the effects of general anesthetics on neuro-apoptosis which may lead to neurodevelopmental impairments or cognitive deficits in later life [10]. However, in the instances of sevoflurane, there exists considerable controversy in the literature when rodents are used as experimental models. Specifically, anesthetics have been reported to negatively impact different behavioural repertoires or memory sub-types such as spatial recognition [24,25,54] and hippocampal-dependent memory [26,55]. On the other hand, some studies have reported that there were no observed effects on any of the above parameters [27,28,56], while others have demonstrated that sevoflurane exposure improves learning and memory [29,57]. Interestingly, the studies that reported no impact, or instead an improvement in an animal’s cognitive abilities, had infused sevoflurane with pure oxygen [27,28,29,56]. In contrast, the studies that reported cognitive deficits had used 30–60% oxygen mixtures [24,25,26,55]. In our study, we infused 1.6% sevoflurane in a 75% oxygen mixture for 1 h either on day one or two and then measured learning and memory at later time points. Perhaps the oxygen concentration administered at the time of sevoflurane exposure may have influenced the apparent effects seen on learning and memory with hyper-oxygenation potentially masking the harmful effects of the anesthetic. Consistent with this postulate are the studies that have shown that increasing oxygen concentration after traumatic brain injury may activate anti-apoptotic and pro-neurogenesis mechanisms [58,59].

An important finding of our research is that in addition to the most commonly used parameters for open field tests, we also developed heatmaps to better understand the exploratory behaviour of animals around the entire field of study and not just in one confined area. While changes in animal behaviour following sevoflurane exposure using open field testing have not been reported before, this paradigm revealed a novel aspect of learning and memory deficit. Specifically, we discovered that the neonatal animals exposed twice to sevoflurane were “hesitant” and “reluctant” to leave certain areas within the box, a behavioural repertoire that we did not observe in the control animals. This is an important observation that suggests that notwithstanding their normal locomotor behaviour and good physical health, the sevoflurane-exposed animals appeared reluctant to explore other regions of their abode or to take the initiative to venture out of their “comfort zone” [49]. These observations thus underscore the importance of future studies in humans where young children that had previously been subjected during early development to multiple anesthetic exposures must be examined for such a behavioural phenotype.

For the Morris Water Maze test, the main indicators of behavioural repertoire exploited in previous studies that were designed to determine the impact of anesthetics on learning deficits have been the differences in latency to reach the platform between days 1 and 5 or the differences in latency on day 5 compared to the control. Interestingly, in the present study, the tracings of the swimming patterns revealed some subtle anomalies that had not been appreciated before. Specifically, we noted a more hesitant, anxious, erratic, and indecisive pattern in those animals that had been exposed to sevoflurane. Several aspects of this behavioural phenotype are analogous to those seen in the open field testing, and thus further expose a novel and unique impact of early life sevoflurane exposure on learning and memory. Additionally, we observed several other unique differences in the learning capabilities when the sevoflurane-exposed animals were tested using the mixed novel object recognition paradigm. Specifically, we discovered that the animals exposed to sevoflurane had their recognition memory compromised as they were not able to recognize that a novel object had been introduced in the box in replacement of the familiar object. However, the animals were indeed capable of recognizing a change in its location, further underscoring that their spatial memory was not impacted. Nevertheless, studies that used a sevoflurane infusion in pure oxygen have not seen this type of memory altered [27,56], which raises the possibility that our results might have been different if we had administered sevoflurane in pure oxygen instead of 75% O_2_. Nevertheless, we have made several novel observations that indicate that neonatal sevoflurane exposure results in long-term and permanent deficits in several aspects of learning and memory in our rodent model. Moreover, these effects go beyond the cognitive functional parameters that have previously been used to examine long-term anesthetic-induced effects on learning and memory in children [11,13,14,60].

## 5. Conclusions

The present study thus underscores the efficacy of DEX pre-treatment in rescuing sevoflurane-induced detrimental effects on neuronal health both in vitro as well as in vivo. We have demonstrated that dexmedetomidine pre-treatment of cultured cortical neurons reversed sevoflurane-induced cytotoxicity and promoted mitochondrial health. Additionally, dexmedetomidine pre-treatment mitigated the negative effects of neonatal sevoflurane exposure on learning and memory in adult rats. These findings will likely have important clinical implications and highlight the need for additional studies, particularly in humans, to further examine and characterize the effects of anesthetics and protective agents on the developing brain.

## Figures and Tables

**Figure 1 biomedicines-11-00391-f001:**
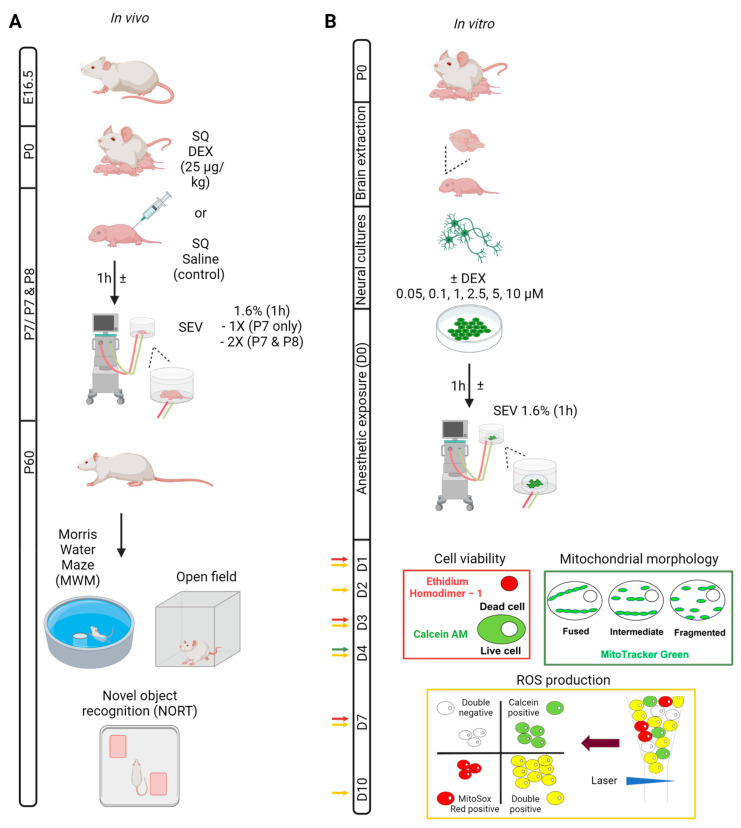
Schematic representation of experimental procedures. (**A**) In vivo procedure: pregnant rat arrives at E16.5 and pregnancy is monitored until P0, where pups are born. Pups are monitored until P7 when they are exposed to subcutaneous dexmedetomidine (using an equivalent volume of 0.1 mL of saline for the control animals) followed by 1.6% sevoflurane exposure for 1 h (single (P7)) or double exposure (P7 and P8) to analyze the effect on learning and memory in the adult animals (P60) using a batch of behavioural tests: (1) Morris Water Maze to assess spatial memory; (2) open field test to analyze anxiety, locomotor behaviour and willingness to explore; and (3) novel object recognition memory test to assess both recognition and spatial memory. (**B**) In vitro procedure: Pups are collected and sacrificed at P0, where the cortices are collected and used for the generation of primary neural cultures. These cortical neurons are then pre-treated with increasing concentrations of dexmedetomidine (media only for the controls) and then exposed to sevoflurane 1.6% for 1 h to determine their effects on different cellular parameters: (1) Live/dead assay using ethidium homodimer 1 as a dead cell marker and calcein AM as the live cell marker; (2) mitochondrial morphology assessed by analyzing fragmentation levels of mitochondrial networks stained with MitoTracker Green; and (3) ROS production assessed by the quantification of mean fluorescence intensity of cells double stained with calcein AM and MitoSox Red using flow cytometry.

**Figure 2 biomedicines-11-00391-f002:**
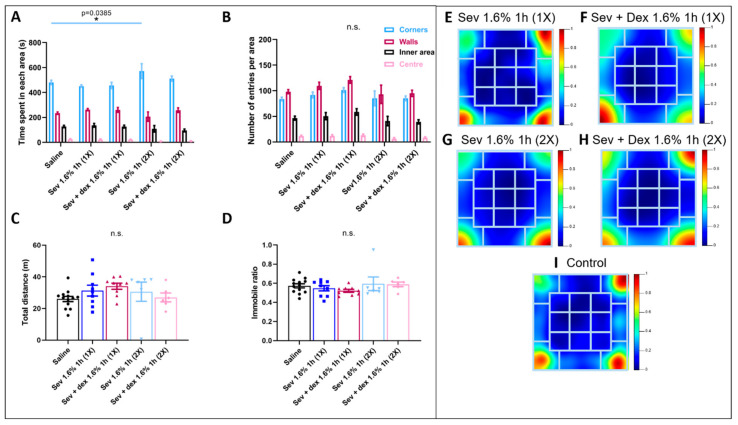
DEX pre-treatment decreased anxiety and increased willingness to explore during open field testing. Representation of (**A**) time spent in each area, (**B**) number of entries in each area, (**C**) total distance travelled, (**D**) immobile ratio, and heatmaps representing the average time spent in each area of the box in (**E**) sevoflurane 1.6% 1 h (1×) animals, (**F**) Sev + Dex 1.6% 1 h (1×) animals, (**G**) sevoflurane 1.6% 1 h (2×) animals, (**H**) Sev + Dex 1.6% 1 h (2×) animals, and (**I**) control animals (saline) (dark blue—less time, dark red—more time). Bars are mean ± SEM (**A**) F (12,152) = 2.440, * *p* = 0.0385 and (**B**) F (12,152) = 0.6265, *p* = 0.8173 using two-way ANOVA with Tukey’s post hoc analysis for multiple comparisons and (**C**) F (4,38) = 1.388, *p* = 0.2565 and (**D**) F (4,36) = 1.022, *p* = 0.4089 using one-way ANOVA with Dunnett’s post hoc analysis for multiple comparisons. *n* = 13 for control, *n* = 9 for Sev and Sev + Dex 1.6% 1 h (1×) and *n* = 6 for Sev and Sev + Dex 1.6% 1 h (2×).

**Figure 3 biomedicines-11-00391-f003:**
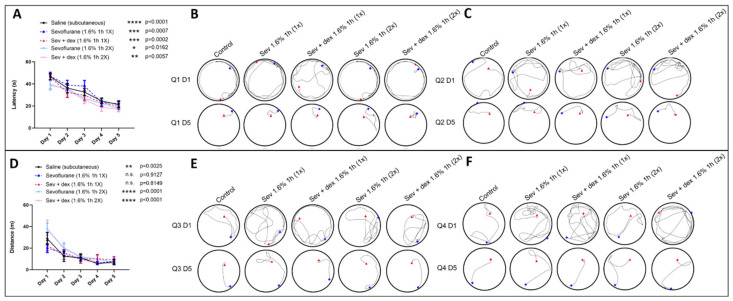
DEX pre-treatment resulted in less anxious, erratic, and indecisive swimming patterns. (**A**) Average latency and (**D**) average swimming distance between the four quadrants needed per day to reach the hidden platform. Swimming tracing patterns for days 1 and 5 in quadrants (**B**) 1, (**C**) 2, (**E**) 3 and (**F**) 4. Control (animals injected with saline). Bars indicate mean ± SEM (**A**) F (16,220) = 0.3397, **** *p* < 0.0001, and (**D**) F (16,220) = 0.9185, **** *p* < 0.0001 using two-way ANOVA with Tukey’s post hoc analysis for multiple comparisons comparing the latency and distance for each group on days 1 and 5. *n* = 13 and *n* = 9 for control and experimental groups, respectively. n.s. means not specified.

**Figure 4 biomedicines-11-00391-f004:**
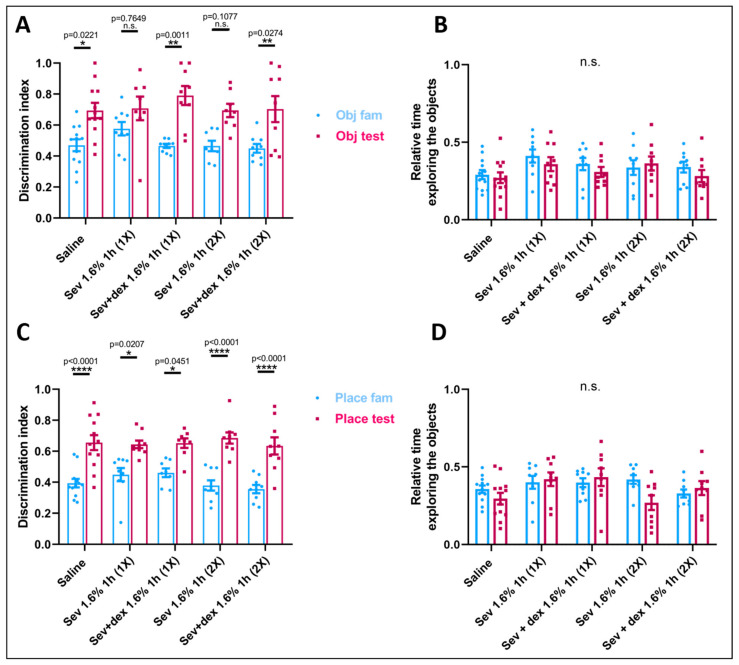
Pre-treatment with subcutaneous DEX was able to prevent recognition memory deficits. (**A**) Discrimination index calculated as the time spent with the novel object (TN) divided by the total time spent with both objects (TN + TF) and (**B**) relative time interacting with both objects (TN+TF)/(duration of the phase). (**C**) Discrimination index calculated as the time spent with the object located in a different place (TN) divided by the total time spent with both objects (TN + TF) and (**D**) relative time interacting with both objects (TN + TF)/(duration of the phase). Bars indicate mean ± SEM (**A**) F (4,85) = 0.4862, ** *p* = 0.0011, ** *p* = 0.0274, and * *p* = 0.0221, (**B**) F (4,94) = 0.7172, *p* = 0.5823 (**C**) F (4,86) = 1.253, **** *p* < 0.0001, (**D**) and F (4,94) = 1.867, *p* = 0.1226 using two-way ANOVA with Tukey’s post hoc analysis for multiple comparisons. *n* = 12 and *n* = 9 for control and experimental groups, respectively.

**Figure 5 biomedicines-11-00391-f005:**
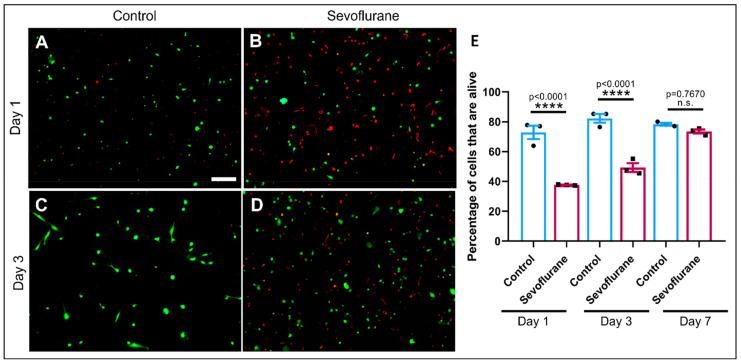
Sevoflurane exposure resulted in increased cell death. Representative live-fluorescent images of neuronal cultures stained with calcein-AM (green) and ethidium homodimer-1 (red) for live and dead cells, respectively. (**A**) Control cells, (**B**) cells exposed to sevoflurane 1.6% (1 h) on day 1 after sevoflurane exposure and (**C**) control cells, (**D**) cells exposed to sevoflurane 1.6% (1 h) on day 3 after sevoflurane exposure. Scale bar indicates 100 μm. (**E**) Percentage of cells that are alive on day 1: Control = 72.87 ± 4.515, Sevoflurane = 37.66 ± 0.2028; day 3: Control = 82.3 ± 2.928, Sevoflurane = 49.25 ± 2.988; and day 7: Control = 78.33 ± 0.8819, Sevoflurane = 73.47 ± 1.316. Values are mean ± SEM F (5, 12) = 47.58, **** *p* < 0.0001 using one-way ANOVA with Dunnett’s post hoc analysis for multiple comparisons. Bars indicate ± SEM, *n* = 3 dishes per condition, 15–20 images per plate.

**Figure 6 biomedicines-11-00391-f006:**
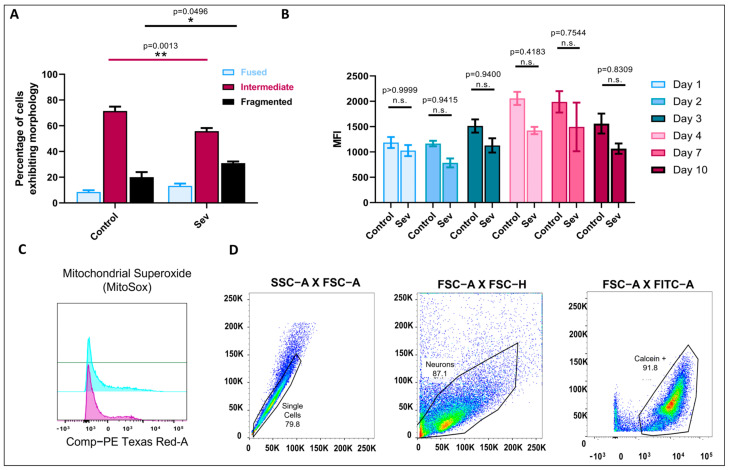
Sevoflurane exposure resulted in increased fragmented mitochondria. (**A**) Mitochondrial morphology quantification on day 4 post-anesthetic exposure comparing control versus sevoflurane-exposed cells. Bars indicate ± SEM, *n* = 8 dishes per condition, 15–20 images per plate. (**B**) Quantification of MFI of mitochondrial ROS production by control and sevoflurane-exposed cells on days 1, 2, 3, 4, 7, and 10. (**C**) Representative mean fluorescence intensity (MFI) of mitochondrial superoxide production by neurons exposed to sevoflurane compared to controls (untreated cells). (**D**) Gating strategy to select living neurons stained with Calcein AM. Bars are mean ± SEM (**A**) F (2,42) = 14.57, ** *p* = 0.0013 and * *p* = 0.496 using one-way ANOVA with Dunnett’s post hoc analysis for multiple comparisons in (**A**) and in (**B**) F (5,35) = 0.3706, *p* = 0.8653 by two-way ANOVA with Tukey’s post hoc analysis for multiple comparisons in (**D**). Cells extracted from *n* = 3–4 individual brains per condition.

**Figure 7 biomedicines-11-00391-f007:**
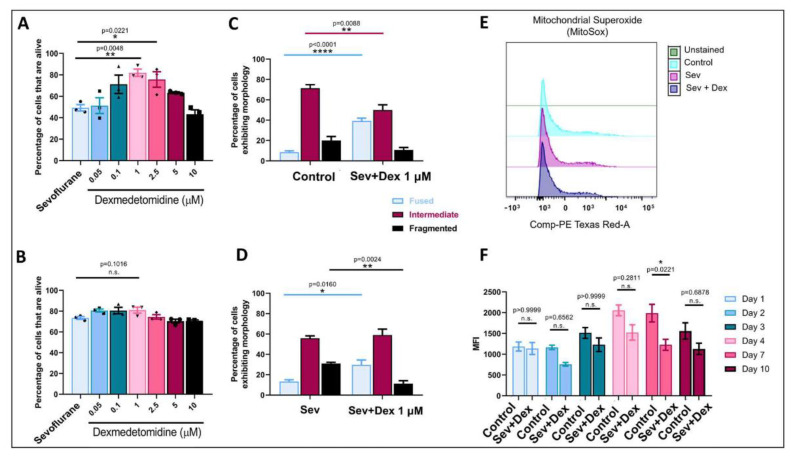
Dexmedetomidine pre-treatment was able to rescue sevoflurane-induced cell death, reverse mitochondrial fragmentation, and promote a decrease in ROS production. (**A**) Percentage of cells that are alive on day 3 (**B**) and day 7. Concentrations of dexmedetomidine expressed in μM. Bars indicate ± SEM *n* = 3 dishes per condition, 15–20 images per plate. Mitochondrial morphology quantification on day 4 post-anesthetic exposure comparing (**C**) control (untreated cells) versus DEX-pre-treated cells and (**D**) sevoflurane-exposed cells versus Sev + Dex cells. Bars indicate ± SEM *n* = 7–8 dishes per condition, 15–20 neurons imaged per plate. (**E**) Representative MFI of mitochondrial superoxide production by neurons exposed to sevoflurane and DEX-pre-treated cells exposed to sevoflurane compared to controls (**F**) Quantification of MFI of mitochondrial ROS production in Sev + Dex cells on days 1, 2, 3, 4, 7, and 10 compared to untreated cells (control). In (**A**), bars are mean ± SEM F (6,14) = 6.951, ** *p* = 0.0048 and * *p* = 0.0221, (**B**) F (6,14) = 5.220, *p* = 0.0052, (**C**) F (2,39) = 11.43, **** *p* < 0.0001, ** *p* = 0.0088 and (**D**) F (2,39) = 14.68, ** *p* = 0.0024, * *p* = 0.0160 using one-way ANOVA with Dunnett’s post hoc analysis for multiple comparisons in (**F**) F (5,35) = 1.442, * *p* = 0.0221 using two-way ANOVA with Tukey’s post hoc analysis for multiple comparisons. Cells extracted from *n* = 3–4 individual brains per condition.

## Data Availability

The data presented in this study are available on request from the corresponding author.

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
