# Peer review of "Dexmedetomidine Pre-Treatment of Neonatal Rats Prevents Sevoflurane-Induced Deficits in Learning and Memory in the Adult Animals"

_biomedicines, 2023, doi:10.3390/biomedicines11020391_

Round 1

Reviewer 1 Report

This long manuscript that explores sevoflurane and dexmedetomidine effects includes results of experiments in rat pups (with behavior measured at 60 days) and in cultured cells obtained from postnatal day zero rats.  Although complex, as a whole, the manuscript is logical and results are well presented.  Some suggestions to make this manuscript easier for the reader:

1.  Increase the font size for labels on figures 1, 2, 3, 6, 7.  The font size labeling bars of Figure 5 was sufficient for easy reading, but it was a struggle to read the others and sometimes required a magnifying glass.

2.  Sevoflurane was given by inhalation and DEX was given subcutaneously in the experiments described.  What prompted subcutaneous administration?  This drug is usually given IV to people and IV or IM to veterinary patients.  

3.  Rewriting of the first sentence of the paragraph that beings on line 284 of page 7 would help the reader--perhaps 'either' rather than 'neither' should be used.

4.  No significant differences were seen in many experiments, but some groups of animals showed differences.  The authors attempt to note this still led to this reader's perception that the significant results were predominant.

5.  The Discussion did compare this study with previous studies and noted important differences.

Author Response

We are grateful to the reviewers for their suggestions and constructive comments; most issues raised were fair and are now addressed below on a point-to-point basis. This paper is now revised in light of the referees' suggestions.  

Reviewers’ comments

Reviewer 1

This long manuscript that explores sevoflurane and dexmedetomidine effects includes results of experiments in rat pups (with behavior measured at 60 days) and in cultured cells obtained from postnatal day zero rats.  Although complex, as a whole, the manuscript is logical and results are well presented.  Some suggestions to make this manuscript easier for the reader:

  1. Increase the font size for labels on figures 1, 2, 3, 6, 7.  The font size labeling bars of Figure 5 was sufficient for easy reading, but it was a struggle to read the others and sometimes required a magnifying glass.

We have modified figures 1, 6, and 7 but have left the original figures 2 and 3 as they were because the suggested font style, will not match with the rest of the figures. Moreover, these inconsistencies in format size will make figures aesthetically unpleasing, and will also be inconsistent with the stylistic requirements of the journal.

  1. Sevoflurane was given by inhalation and DEX was given subcutaneously in the experiments described.  What prompted subcutaneous administration?  This drug is usually given IV to people and IV or IM to veterinary patients.  

Another notable difference between our study and the previously published work is that instead of the conventional intraperitoneal (IP) delivery method for DEX, we used the subcutaneous mode of DEX pre-treatment. Although the common method of giving anesthetics to rodents is IP, intravenous administration (IV) is preferred clinically1. We chose to use DEX as a subcutaneous treatment as IV DEX administration has been shown to cause significant changes in heart rate and blood pressure2, whereas subcutaneous administration only causes a smaller decline in these parameters thus reducing the risk of hemodynamic depression2,3. We did not find any significant differences in respiratory rate or SpO2 in our DEX-injected animals in comparison to their control counterparts. When administered subcutaneously, we observed a significant rescue effect by DEX on all parameters of sevoflurane-induced toxicity.

  1. Rewriting of the first sentence of the paragraph that beings on line 284 of page 7 would help the reader--perhaps 'either' rather than 'neither' should be used.

Thanks for pointing that out. It has been fixed now.

  1. No significant differences were seen in many experiments, but some groups of animals showed differences.  The authors attempt to note this still led to this reader's perception that the significant results were predominant.

The point we wanted to make is that during our study, we have noticed subtle changes, which have either been not reported or disregarded previously. For instance, the trajectories during the MWM led us to conclude that sevoflurane did not cause spatial memory deficits at this subclinical dose as measured by the latency reported in other studies but instead, we found that these animals exhibited an erratic swim pattern - swimming faster in anticipation of locating their platform. Specifically, the animals appeared “anxious” to find the platform (intense swimming behavior), albeit not knowing initially where it might have been located. In contrast, the control animals had no difficulty “remembering” where the potential platform would likely be, and they swam directly toward their desired target.

Furthermore, in our study, we used subcutaneous dexmedetomidine instead of intraperitoneal and still we found that this route of administration managed to provide neuroprotective effects, therefore, highlighting the significance of this study both at the subtle changes in behavioral patterns not shown before plus a different route of administration for DEX.

References:

  1. Turner, P. V., Brabb, T., Pekow, C. & Vasbinder, M. A. Administration of Substances to Laboratory Animals: Routes of Administration and Factors to Consider. J. Am. Assoc. Lab. Anim. Sci. JAALAS 50, 600–613 (2011).
  2. Uusalo, P. et al. Subcutaneously administered dexmedetomidine is efficiently absorbed and is associated with attenuated cardiovascular effects in healthy volunteers. Eur. J. Clin. Pharmacol. 74, 1047–1054 (2018).
  3. Álvarez-Betancourt, A. E., Sánchez-Hernández, E., Guadalupe  López-González, B. & Armando  Rodríguez-Moreno, Ó. Subcutaneous dexmedetomidine. Is it useful in the perioperative of the pediatric patient? Rev. Mex. Anestesiol. 43, 16–22 (2020).

Reviewer 2 Report

The study demonstrates that exposure to sub-clinically used concentrations of inhalation anesthetic sevoflurane results in significant learning and memory deficits in adult animals, which involve neuronal cell death likely mediated through perturbed mitochondrial structure and function. The study also found that these deficits and cell death can be prevented by pre-treatment with the anesthetic adjuvant Dexmedetomidine. Expeirmental are well reformed, results are clearly explained, however, to bring more clarity to repeat experiments and clarify the results and extrapolation to real clinics application, there are relevant gaps, that needs to be addressed. My comments are appended below

Introduction

Sentence refereeing to synaptic plasticity in line 56-58 needs reference, I suggest citing an important seminal work https://doi.org/10.1371/journal.pone.0193867 along with sentence ´ In most instances, the potential extrinsic interrupters of brain network connectivity could be avoided, or the shortcomings managed through synaptic plasticity involving compensatory mechanisms ´

Materials Method

Please "from different breeding groups" or "different female animals" shall be used to refer to using ´different mothers´ in animal experiments.

In Materials and method, please add catalogue number, supplier, country/region of all chemicals/reagents (e.g. Cat # missing sevoflurane  or DEX used), kit/assay catalogue no (e.g. live/dead assay), supplier, instrument with name details (e.g. Healthcare Gas Analyzer), model number, version etc. This is mandatory as Committee on Publication Ethics (COPE) guidelines adopted by Wiley and other journals to add full details of material to replicate the experiment independently in any lab who wish to cross verify or reproduce the results. 

Provide a list of all abbreviations used in this manuscript as some of them (e.g. ARRIVE guidelines) are used without padded full forms, which makes it hard to follow.

Results and discussion

Figure 1: Schematic representation of experimental procedures described, it is not easy to understand labels in ROS Box. Provide a good resolution image.

What’s gating criteria authors adopted for the FACS scatter plot analysis, mention in experimental sections.

What were the specific learning and memory deficits observed in adult animals as a result of sevoflurane exposure? Provide in tabular form to comprehend it better. How does sevoflurane exposure lead to neuronal cell death, explain mechanism in discussion of results? Are these learning and memory deficits observed in human?

Cite a latest report ttps://doi.org/10.3390/cells11182801 published in MDPI Journal to support the statement along line 545-547 `These data demonstrate that notwithstanding its effects on mitochondrial morphology, neuronal sevoflurane exposure does not affect ROS production and as such, this may not be the underlying cause for the observed anesthetic-induced cell death.`

How does pre-treatment with Dexmedetomidine prevent the negative effects of sevoflurane exposure?

Can this study result be extrapolated to other inhaled anesthetics? Are there any long-term effects of sevoflurane exposure on learning and memory? Add to discussion points.

Are there any alternative anesthetics or adjuvants that can be used to mitigate the negative effects of sevoflurane exposure?

What are the implications of this study for clinical practice and future research on anesthetic agents add in conclusion section?

Author Response

We are grateful to the reviewers for their suggestions and constructive comments; most issues raised were fair and are now addressed below on a point to point basis. This paper is now revised in light of the referees' suggestions.  

Reviewers’ comments

Reviewer 2

The study demonstrates that exposure to sub-clinically used concentrations of inhalation anesthetic sevoflurane results in significant learning and memory deficits in adult animals, which involve neuronal cell death likely mediated through perturbed mitochondrial structure and function. The study also found that these deficits and cell death can be prevented by pre-treatment with the anesthetic adjuvant Dexmedetomidine. Expeirmental are well reformed, results are clearly explained, however, to bring more clarity to repeat experiments and clarify the results and extrapolation to real clinics application, there are relevant gaps, that needs to be addressed. My comments are appended below:

  1. Introduction

Sentence refereeing to synaptic plasticity in line 56-58 needs reference, I suggest citing an important seminal work https://doi.org/10.1371/journal.pone.0193867 along with sentence ´ In most instances, the potential extrinsic interrupters of brain network connectivity could be avoided, or the shortcomings managed through synaptic plasticity involving compensatory mechanisms ´

Thanks for the reference suggestion, it has been added now.

  1. Materials Method

2.1. Please "from different breeding groups" or "different female animals" shall be used to refer to using ´different mothers´ in animal experiments.

It has been corrected now.

2.2. In Materials and method, please add catalogue number, supplier, country/region of all chemicals/reagents (e.g. Cat # missing sevoflurane  or DEX used), kit/assay catalogue no (e.g. live/dead assay), supplier, instrument with name details (e.g. Healthcare Gas Analyzer), model number, version etc. This is mandatory as Committee on Publication Ethics (COPE) guidelines adopted by Wiley and other journals to add full details of material to replicate the experiment independently in any lab who wish to cross verify or reproduce the results. 

Thanks for pointing out the oversight. It has now been added.

2.3. Provide a list of all abbreviations used in this manuscript as some of them (e.g. ARRIVE guidelines) are used without padded full forms, which makes it hard to follow.

The list of abbreviations was not part of this journal format but we will add it if the editor finds it necessary.

  1. Results and discussion

3.1. Figure 1: Schematic representation of experimental procedures described, it is not easy to understand labels in ROS Box. Provide a good resolution image.

It has been fixed now.

3.2. What’s gating criteria authors adopted for the FACS scatter plot analysis, mention in experimental sections.

The gating criteria have now been added to the 2.11. Reactive oxygen species (ROS) production using flow-cytometry section

3.3.What were the specific learning and memory deficits observed in adult animals as a result of sevoflurane exposure? Provide in tabular form to comprehend it better.

Please see the table below for clarification.

Test

Behavioral aspect

Sevoflurane alone effect

DEX pre-treatment effect

Open field

Locomotor skills

Unaffected

Unaffected

Exploratory mechanisms

Increased corner exploration

Decreased corner exploration

Anxiety

Increased anxiety-like behaviors (confinement to sheltered areas)

Decreased anxiety-like behaviors (confinement to sheltered areas)

MWM

Latency (learning how to find the platform)

Unaffected, learning how to find the platform over the course of 5 days

Unaffected, learning how to find the platform over the course of 5 days

Distance to find the platform (spatial memory)

Groups Sev and Sev+dex (2X) had longer initial distances to find the platform on day 1

Trajectory (anxiety)

Erratic trajectories, intense swimming behavior (anxiety-like behaviors, unsure where the platform was)

Improvement in trajectory patterns, straighter forward, animals surer where the platform was, reduced anxiety-like behaviors

Novel object test

Recognition memory

Significantly affected

Significantly improved

Novel place test

Spatial memory

Unaffected

Unaffected

3.4.a. How does sevoflurane exposure lead to neuronal cell death, explain mechanism in discussion of results?

With the data from our study, we can only speculate about the potential mechanism involving mitochondria and ROS production.

3.4.b. Are these learning and memory deficits observed in human?

The parameters studied in kids who had been exposed to sevoflurane involved the analysis of other behavioral parameters that cannot generally be assessed in the animals (e.g. language, verbal reasoning, etc.) but the human studies did not suggest any apparent behavioral deficits. A few studies have, however, eluded some aspects of learning and memory deficit in young children who were exposed to anesthetics either once or twice in their lives. The evidence however remains equivocal but those studies did compel the Health Authorities to require labeling all anesthetics as potential risk factors – especially when given to pregnant mothers and young children. The evidence provided in the present study will now offer a novel paradigm for clinicians to examine in young children who are exposed to anesthetics.  

3.5. Cite a latest report ttps://doi.org/10.3390/cells11182801 published in MDPI Journal to support the statement along line 545-547 `These data demonstrate that notwithstanding its effects on mitochondrial morphology, neuronal sevoflurane exposure does not affect ROS production and as such, this may not be the underlying cause for the observed anesthetic-induced cell death.`

We would have loved to cite this paper but the provided reference has nothing to do with the issue raised nor the data presented in our study. The paper provided is titled (Micropatterned Neurovascular Interface to Mimic the Blood–Brain Barrier’s Neurophysiology and Micromechanical Function: A BBB-on-CHIP Model which has nothing to do with our statement).

3.6. How does pre-treatment with Dexmedetomidine prevent the negative effects of sevoflurane exposure?

From our study, we anticipate that there is an underlying mechanism involving mitochondrial protection as DEX causes mitochondrial hyperfusion and ROS production reduction.

3.7. Can this study result be extrapolated to other inhaled anesthetics?

In one of our previous papers (doi: 10.1038/s41598-021-84168-y) we showed that desflurane (another commonly used inhaled anesthetic) similarly caused increased mitochondrial fragmentation as sevoflurane so it is likely that dexmedetomidine could be used as a pre-treatment to prevent the mitochondrial fragmentation in a similar way.

3.8. Are there any long-term effects of sevoflurane exposure on learning and memory? Add to discussion points.

Please refer to 656-664 to understand what has been previously done with sevoflurane on animal studies and the controversial results. We explained in detail our findings on behavioral parameters starting in line 675.

3.9. Are there any alternative anesthetics or adjuvants that can be used to mitigate the negative effects of sevoflurane exposure? What are the implications of this study for clinical practice and future research on anesthetic agents add in conclusion section?

With this study, we are setting up the bases to examine further the mechanisms by which sevoflurane causes neuronal toxicity during the administration in the neonatal stage and how this effect can last even in the adult animals. With our data, we can only speculate that potential drugs with an effect on mitochondria might be used to prevent toxicity where mitochondria are affected.